

# Viability of rotation sensing using phonon interferometry in Bose-Einstein condensates

Charles W. Woffinden[1,2*°], Andrew J. Groszek[1,2,3°], Guillaume Gauthier[1,2],
Bradley J. Mommers[1,2], Michael W. J. Bromley[1,4], Simon A. Haine[5],
Halina Rubinsztein-Dunlop[1,2], Matthew J. Davis[1,2,3],
Tyler W. Neely[1,2] and Mark Baker[1,2,6]

**1** School of Mathematics and Physics,
University of Queensland, St. Lucia, QLD 4072, Australia
**2** Australian Research Council Centre of Excellence for Engineered Quantum Systems, School
of Mathematics and Physics, University of Queensland, St. Lucia, QLD 4072, Australia
**3** Australian Research Council Centre in Future Low-Energy Electronics Technologies, School
of Mathematics and Physics, University of Queensland, St. Lucia, QLD 4072, Australia
**4** School of Sciences, University of Southern Queensland, Toowoomba, QLD 4350, Australia
**5** Department of Quantum Science and Technology, Research School of Physics and
Engineering, The Australian National University, Canberra, ACT 2601, Australia
**6** Quantum Technologies Group, Sensors and Effectors Division,
Defence Science & Technology Group

⋆ c.woffinden@uq.edu.au

## Abstract

We demonstrate the use of a ring-shaped Bose-Einstein condensate as a rotation sensor by measuring the interference between two counter-propagating phonon modes imprinted azimuthally around the ring. We observe rapid decay of the excitations, quantified by quality factors of at most $Q \approx 27$. We numerically model our experiment using the c-field methodology, allowing us to estimate the parameters that maximise the performance of our sensor. We explore the damping mechanisms underlying the observed phonon decay, and identify two distinct Landau scattering processes that each dominate at different driving amplitudes and temperatures. Our simulations reveal that $Q$ is limited by strong damping of phonons even in the zero temperature limit. We perform an experimental proof-of-principle rotation measurement using persistent currents imprinted around the ring. We demonstrate a rotation sensitivity of up to $\Delta\Omega \approx 0.3 \, \mathrm{rad \, s^{-1}}$ from a single image, with a theoretically achievable value of $\Delta\Omega \approx 0.04 \, \mathrm{rad \, s^{-1}}$ in the atomic shot-noise limit. This is a significant improvement over the shot-noise-limited $\Delta\Omega \approx 1 \, \mathrm{rad \, s^{-1}}$ sensitivity obtained by Marti et al. [1] for a similar setup.

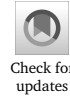

---

° These authors contributed equally to the development of this work.



# Contents

# 1 Introduction

High precision rotation sensing is critical for applications such as inertial navigation [2], geodesy [3], and tests of fundamental physical theories such as general relativity [4]. One mature technology used for rotation sensing is the hemispherical resonator gyroscope (HRG) [2,5], which relies on standing mechanical waves generated around a hemispherical body akin to those produced by tapping the rim of a wine glass. When the hemispherical body is rotated, the standing waves precess around the circumference due to the Coriolis force. This precession is proportional to the rotation rate of the hemisphere and can thus be used to measure the rotation rate. HRGs are used for rotation sensing in applications where high precision and reliability are required, such as on board the James Webb Space Telecope [6]. HRG excitations are robust to damping, quantified by high quality factors of $Q \sim 10^7$. However, their signal is susceptible to drift if used over long periods, and they only provide a measure of relative—rather than absolute—rotation.

In recent decades, cold atoms have emerged as an attractive alternative to traditional optical techniques for performing high precision interferometry. The associated atomic wavelengths and velocities make it possible to realise extremely high precision interferometric devices such as gravimeters, gradiometers, and gyroscopes [7]. For rotation sensing, 'free-space' atom interferometry configurations have proved successful, where the atoms are in free-fall for the duration of the interrogation [8–12]. However, such setups require a large apparatus and are therefore not suitable for many practical sensing applications. More recently, guided or trapped atom interferometers have been demonstrated [1, 13–18]. In these systems, an ultracold gas or superfluid Bose-Einstein condensate (BEC) is confined to a geometry, such as a ring trap, using optical or magnetic potentials [19, 20]. These systems are compact, have an improved lifetime, and produce strong signals due to their high atomic densities. However, measurements are susceptible to systematic phase shifts arising from atom–atom interactions and trap imperfections.

Recently, it was proposed that a ring-shaped BEC could be used for rotation sensing [1, 14] using similar operating principles to an HRG. In this scheme, standing waves are formed using the collective excitations (phonons) of the condensate. Because these phonon modes are the lowest energy excitations available [21], they are anticipated to be resistant to losses into other excitation branches, as well as being minimally affected by interactions and trap inhomogeneity. An added advantage is that superfluids are irrotational, and can therefore serve as an absolute frame of reference for rotation sensing with no need for calibration [22]. While Ref. [1] successfully implemented this interferometry scheme, their sensitivity was too low to perform a practical rotation measurement. Furthermore, the phonons were found to decay at a rate much faster than that predicted by theory, severely limiting the potential sensitivity of the system.

Here, we further explore the properties of the phonon-based rotation scheme proposed in Ref. [1] to determine whether such sensors may be of practical use. Primarily, we aim to improve the sensitivity of this device over the implementation of Ref. [1] and perform a measurement of an applied rotation. We experimentally create a quasi-two-dimensional (2D) ring-shaped BEC using a digital micromirror device (DMD) and perform a sinusoidal phase imprint to excite azimuthal phonon modes. We characterise the performance of our interferometer as both the wavelength and the amplitude of the imprinted mode are varied, observing quality factors up to a maximum of $Q \approx 27$. We numerically model our setup using classical field simulations at finite temperature [23], obtaining $Q$ factors that are in good agreement with the experiment. Using the numerics, we perform a detailed exploration of the physical processes responsible for the rapid phonon damping observed in this system. Our analysis reveals a rich variety of decay behaviour resulting from the interplay between two distinct forms of Landau damping [24–26]. By simulating our system in the limit of zero temperature and weak perturbations where these damping mechanisms are minimised, we predict that the quality of our setup could be improved to $Q \sim 150$, but even in this idealised scenario, the performance is many orders of magnitude below modern HRGs for an experimentally realistic fixed area. Both experimentally and numerically, we apply the phonon interferometry technique to directly measure the rotation rate of persistent currents in the ring. Our measurements imply that our experiment could achieve a rotation sensitivity of up to $\Delta\Omega \approx 0.3 \, \mathrm{rad\,s^{-1}}$ from a single measurement.

## 2 Phonon interferometry for rotation sensing

A phonon interferometer functions in much the same way as a hemispherical resonator gyroscope, whereby the beat frequency between counter-propagating excitations can be used

to provide a measurement of the external rotation rate [2, 5]. The superfluid nature of the medium additionally provides a stationary inertial frame of reference (FoR) for the phonons, and in this frame the counter-propagating modes travel at equal and opposite angular velocities. The azimuthal density of the atoms as a function of time $t$ and angle $\theta$ around the ring can therefore be written:

$$n_\theta(\theta, t) = \bar{n}_\theta + \delta n_\theta(\theta, t) = \bar{n}_\theta + \delta n_\theta^+(\theta, t) + \delta n_\theta^-(\theta, t), \tag{1}$$

where $\bar{n}_\theta$ is the ground state density of the ring, $\delta n_\theta$ is the total density perturbation, and $\delta n_\theta^\pm$ are the perturbations due to the anticlockwise (clockwise) propagating modes in the medium. For the case of azimuthal standing waves, the density perturbations have the form [14]

$$\delta n_\theta^\pm(\theta, t) = \mathcal{A} \sin(m\theta \pm \omega_m t), \tag{2}$$

with $\mathcal{A}$ the perturbation amplitude, $m$ the mode number (corresponding to the number of wavelengths around the ring), and $\omega_m$ the corresponding angular frequency. The total density perturbation in the frame of the superfluid is therefore

$$\delta n_\theta(\theta, t) = 2\mathcal{A} \sin(m\theta) \cos(\omega_m t). \tag{3}$$

If an observer is in the laboratory FoR rotating at a rate $\Omega$ relative to the superfluid, the azimuthal co-ordinate from their perspective shifts, $\theta \to \theta - \Omega t$, and hence they will see the standing wave precess at a frequency $m\Omega$:

$$\delta n_\theta^{(\text{lab})}(\theta, t) = 2\mathcal{A} \sin[m(\theta - \Omega t)] \cos(\omega_m t). \tag{4}$$

By measuring the azimuthal density of the condensate, the observer can therefore deduce the rotation rate $\Omega$.

## 3 Experiment

### 3.1 BEC preparation

The BEC experimental setup and preparation process have been described in detail in Ref. [19]. In brief, a BEC of $^{87}$Rb atoms in the $|F = 1, m_F = -1\rangle$ state, formed using a hybrid optical and magnetic trapping technique [27], is tightly trapped in the vertical ($z$) direction using a far red-detuned 1064 nm sheet beam potential with a trapping frequency $\omega_z = 2\pi \times 133$ Hz. A far blue-detuned 532 nm laser is used to create the horizontal trapping potential and is shaped by direct projection from a DMD. By switching the DMD mirrors to an on or off position, arbitrary and dynamic patterning of the BEC at sub-micron resolution can be achieved [19].

We use a ring-shaped potential in the horizontal plane (see Sec. 3.2), with typical atom numbers of $N_{\text{atoms}} \approx 2 \times 10^6$. We expect the system to be in the Thomas–Fermi regime, with a parabolic density profile in the vertical direction of width $\approx 8\,\mu$m. Using the Thomas–Fermi approximation, we estimate the chemical potential to be $\mu \approx 55\,$nK $\times k_{\text{B}}$. Although the system should be fully 3D in a thermodynamic sense (with a true BEC transition), the tight vertical and radial confinements restrict the relevant dynamics to be along the azimuthal direction. Throughout this work, the condensate fraction of the gas is $n_0 \approx 0.8$, which we find to be the highest achievable in this setup. We estimate this value by letting the cloud evolve in time-of-flight and calculating the fraction of atoms contained within respective fits to the condensed and thermal components, as measured from absorption imaging.

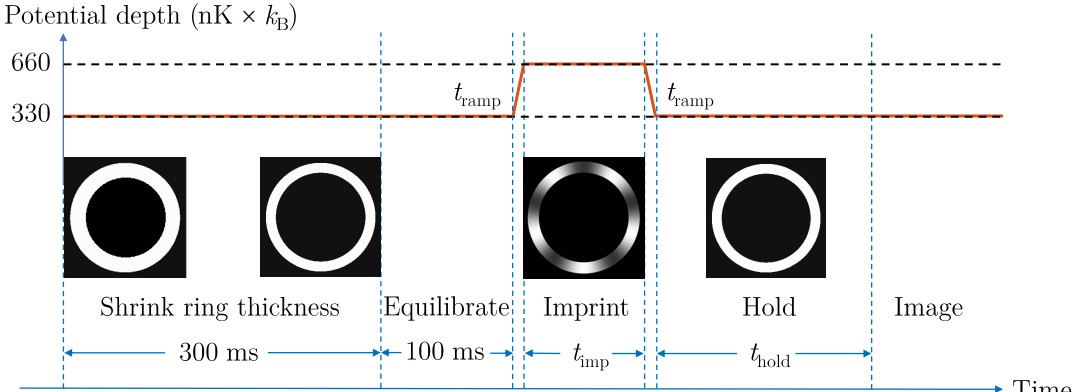

Figure 1: Timing diagram for experimental sequence of ring formation, phonon imprinting, hold time, and imaging. The depth of the trapping potential is indicated with the red line, where $t_{\mathrm{ramp}} = 500\,\mu s$ (time not to scale—see text). Insets show the applied DMD potential for different parts of the sequence, with white areas corresponding to an attractive potential. In the example shown, the imprinted mode number $m^* = 5$.

## 3.2 Experimental sequence

The experimental sequence is summarised in Fig. 1. We initially load the atoms into an annulus with inner and outer radii of $r_{\mathrm{in}} \approx 35\,\mu m$ and $r_{\mathrm{out}} \approx 55\,\mu m$, respectively, and a potential depth of $\approx 330\,nK \times k_B$. The inner radius is then expanded to $r_{\mathrm{in}} \approx 45\,\mu m$, increasing the atomic density and the chemical potential. This lessens the impact of trap imperfections. We find that $10\,\mu m$ is the minimum ring thickness we can achieve that avoids segmentation of the condensate due to light leak through from the DMD. After this step, the atoms are left to equilibrate in the final ring potential for $100\,ms$.

We then imprint an azimuthally varying phase pattern around the ring. This is done by applying a sinusoidal potential $V_{\mathrm{imp}}(\theta)$ for a time $t_{\mathrm{imp}}$, causing the atoms to acquire a phase factor $\exp[i V_{\mathrm{imp}}(\theta) t_{\mathrm{imp}}/\hbar]$. Our imprinting potential takes the form $V_{\mathrm{imp}}(\theta) = V_0 \sin(m^* \theta)$, where $m^*$ is the mode number being excited.[1] This pattern excites two counter-propagating waves, which, due to the superfluid's irrotational nature, must have equal amplitudes. The amplitude $V_0$ is set to $\approx 90\%$ of the trap depth—this value ensures a fast imprint (relative to the response time of the atoms) while also minimising the loss of atoms from the BEC. We apply this imprinting potential via the DMD by adding it to the existing trapping potential.

Immediately prior to the phase imprint, the trapping potential height is doubled to $\approx 660\,nK \times k_B$ over a time $t_{\mathrm{ramp}} = 500\,\mu s$, in order to prevent atoms spilling out of the trap when the imprint is applied. The imprinting potential is switched on for a variable time $60\,\mu s \leq t_{\mathrm{imp}} \leq 300\,\mu s$, with longer imprints generating higher amplitude phase patterns (for significantly longer imprint times, a density perturbation forms, which is the regime explored in Ref. [1]). After this time, the imprinting potential is switched off, and the trap is lowered to its original height over time $t_{\mathrm{ramp}}$, so that stray-light induced losses are minimised during the subsequent evolution. The BEC is then allowed to evolve for a variable hold time $t_{\mathrm{hold}}$ (up to $400\,ms$), after which time the atomic density is measured in situ using Faraday imaging.

---

[1]Throughout this work, we use $m$ for the general azimuthal mode index and denote the imprinted mode $m^*$.

# 4 Numerical modelling

We numerically model the Bose gas using three-dimensional c-field simulations [23], allowing us to approximate the effects of the thermal cloud on the dynamics. In this approach, a complex-valued classical field $\psi(\mathbf{r}, t)$ is used to describe the condensate and a chosen set of macroscopically occupied low-energy excitations. Excitations above the energy cutoff are excluded from the field and are not modelled dynamically. Although c-field models cannot quantitatively capture all aspects of the Bose gas simultaneously [28], they have nonetheless been shown to match experiments when used appropriately [29–31]. Here we use the c-field simulations to provide insight into the mechanisms of the damping behaviour of the system at non-zero temperature, and find that the results are in semi-quantitative agreement with experiments when the condensate fractions are matched.

To sample thermal states, we use the stochastic projected Gross–Pitaevskii Equation [32–34], in which the above-cutoff excitations are treated as a thermal reservoir with temperature $T$ and chemical potential $\mu$, coupled to the field via the dimensionless collision rate $\gamma$. The time evolution is given by:

$$d\psi = \mathcal{P}\left\{-\frac{i}{\hbar}L_{\text{GP}}\psi dt + \frac{\gamma}{\hbar}(\mu - L_{\text{GP}})\psi dt + dW\right\}, \tag{5}$$

where the operator $L_{\text{GP}} = -(\hbar^2/2m_{\text{a}})\nabla^2 + V_{\text{trap}} + g|\psi|^2$, and the complex local white noise $dW$ has correlations $\langle dW^*(\mathbf{r}, t)dW(\mathbf{r}', t)\rangle = (2\gamma k_{\text{B}}T/\hbar)\delta(\mathbf{r} - \mathbf{r}')dt$. The interaction parameter $g = 4\pi\hbar^2 a_{\text{s}}/m_{\text{a}}$, with the $^{87}$Rb scattering length $a_{\text{s}} \approx 5.3$ nm and atomic mass $m_{\text{a}} \approx 1.44 \times 10^{-25}$ kg. The projection operator $\mathcal{P}$ ensures that the field is confined to the classical field region, defined here as the set of three-dimensional plane waves with single-particle energies satisfying $\hbar^2|\mathbf{k}|^2/2m_{\text{a}} < E_{\text{cut}}$, with $E_{\text{cut}}$ the energy cutoff.

We use a trapping potential $V_{\text{trap}}(x, y, z) = V_{xy}(x, y) + V_z(z)$, where $V_z(z) = m_{\text{a}}\omega_z^2 z^2/2$ is a tight harmonic trapping potential, and the planar potential $V_{xy}$ is chosen to be an annulus of the form:

$$V_{xy}(x, y) = \tilde{V}_0\left\{1 + \frac{1}{2}\tanh\left[\frac{2}{\sigma}(r - r_{\text{out}})\right] - \frac{1}{2}\tanh\left[\frac{2}{\sigma}(r - r_{\text{in}})\right]\right\}, \tag{6}$$

with $r = \sqrt{x^2 + y^2}$ the radial co-ordinate. This potential has maximum height $\tilde{V}_0$, inner (outer) radius of $r_{\text{in}}$ ($r_{\text{out}}$), and barrier width set by $\sigma$.

To generate the initial condition for our simulations, we first obtain the ground state of the potential by evolving Eq. (5) with $\gamma = 1$ and $T = 0$. We then set the temperature $T \geq 0$ and continue to evolve in time until thermal equilibrium is reached. Finally, we imprint a phase profile:

$$\psi \to \psi \exp\left[i\Phi_{\text{imp}}\cos(m^*\theta)\right], \tag{7}$$

where $\Phi_{\text{imp}}$ is the imprinted amplitude of the phase profile and $m^*$ is the imprinted mode number. We then evolve this initial state in time using the projected Gross–Pitaevskii equation [23], obtained by setting $\gamma = 0$ in Eq. (5). In this model, the classical field is decoupled from the thermal reservoir, and hence energy and particle number are both conserved under time evolution. The decay of the imprinted phonon therefore results from the internal redistribution of population among the available modes, rather than loss of energy to the reservoir.

In Eq. (5), we use a chemical potential $\mu = 58$ nK $\times k_{\text{B}}$ and (unless otherwise stated) a temperature $T \approx 170$ nK. These values are chosen such that the number of atoms approximately matches the experiment, and the decay times of the imprinted phonons are on the order of those measured experimentally. With these parameters, we find that the condensate fraction

is $n_0 \approx 0.81$,[2] although we note that this parameter should be treated as a self-consistent measure of coherence within the c-field model, and may be different from the experimentally measured value.

The planar potential is chosen to have barrier height $\tilde{V}_0 = 5\mu = 290\,\text{nK} \times k_\text{B}$, wall width $\sigma \approx 2\,\mu\text{m}$, and inner (outer) radius of $r_\text{in} = 45\,\mu\text{m}$ ($r_\text{out} = 55\,\mu\text{m}$). We use a numerical grid of $256 \times 256 \times 24$ points, corresponding to a simulation domain of $132\,\mu\text{m} \times 132\,\mu\text{m} \times 16\,\mu\text{m}$. We numerically evolve Eq. (5) using XMDS2 [36]. A second-order semi-implicit differential equation solver is used when $\gamma > 0$, and a fourth-order Runge–Kutta scheme is used when $\gamma = 0$. The energy cutoff is set to $E_\text{cut} = \hbar^2 k_\text{cut}^2 / 2m_\text{a} \approx 62\,\text{nK} \times k_\text{B}$ (corresponding to wavenumber cutoff $k_\text{cut} \approx 4.7\,\mu\text{m}^{-1}$). The imprinted mode number is varied between $3 \leq m^* \leq 13$, and the amplitude is varied in the range of $0.1\pi \leq \Phi_\text{imp} \leq 8\pi$. We note that although we have used a 3D model for these simulations, we find that 2D simulations with the same atom number and condensate fraction provide quantitatively similar predictions for the decay of the imprinted excitation.

## 5 Analysis

### 5.1 Measurement protocol

For both experimental and numerical data, we analyse the evolution of the azimuthal excitations by measuring the atomic density. Following the method presented in Ref. [1], we perform a Fourier transform of the density to obtain the complex Fourier amplitude $A_m$ of the $m^\text{th}$ azimuthal mode:

$$A_m(t) = \frac{\iint n_\text{2D}(\mathbf{r}, t)e^{-im\theta}\,\mathrm{d}\mathbf{r}}{\iint n_\text{2D}(\mathbf{r}, t)\,\mathrm{d}\mathbf{r}} = A_m^{(R)}(t) + iA_m^{(I)}(t), \tag{8}$$

where $n_\text{2D}(\mathbf{r}, t)$ is the two-dimensional ($z$-integrated) atomic density profile at time $t$, and $A_m^{(\nu)}$ are the real ($\nu = R$) and imaginary ($\nu = I$) components of the amplitude. Throughout this work, $A_m(t)$ is normalised by the atom number, such that $0 \leq |A_m(t)| \leq 1$.

Treating the radial density profile as time-independent, the two-dimensional density can be expressed as a product $n_\text{2D}(\mathbf{r}, t) = n_r(r)n_\theta(\theta, t)$. Substituting this into Eq. (8) and using $n_\theta$ given by Eqs. (1) and (4), the amplitude of the excited phonon mode $m = m^*$ as measured from the lab frame will have the form

$$A_{m^*}(t) = i\alpha(t)\sin(\omega_{m^*}t + \phi_0)e^{im^*\Omega t}. \tag{9}$$

For generality, we have included a phase offset $\phi_0$ and a time-varying envelope $\alpha(t)$ to account for any damping. As in previous works [1, 31, 37], we find that the envelope $\alpha(t)$ is well described by exponential decay in most cases. We will therefore assume

$$\alpha(t) = \alpha_0 e^{-t/\tau}, \tag{10}$$

throughout most of this work. Here, $\tau$ is the decay constant and $0 \leq \alpha_0 \leq 1$ is the imprint amplitude, controlled by either $t_\text{imp}$ (in the experiment) or $\Phi_\text{imp}$ (in the numerics). In the case $|\Omega| > 0$ considered in Sec. 7, we extract the rotation rate by measuring the angle $\theta(t) \equiv \arg\{A_{m^*}(t)\}$, which grows linearly in time as $\theta(t) \propto m^*\Omega t$.

---

[2]Determined from the largest eigenvalue of the one-body density matrix, in accordance with the Penrose–Onsager criterion [23, 35].

## 5.2 Quality factor of the phonon interferometer

The quality or $Q$ factor of a resonator is typically defined as [38]

$$Q = \frac{\omega_m \tau}{2}, \tag{11}$$

where $\omega_m$ is the angular frequency of the excitation, and $\tau$ is its lifetime. The frequency of the $m^{\text{th}}$ mode can be approximated as $\omega_m \approx m\omega_0$, where $\omega_0 \approx c_s/R$ is the fundamental frequency of the ring, determined by the speed of sound $c_s = \sqrt{\mu/m_a}$ and the ring radius $R$.

In principle, the quality may be improved by either increasing $\omega_m$ or $\tau$. However, in this system it has previously been demonstrated that $\tau \propto m^{-1}$ [1], and thus the product $\omega_m \tau$ should remain constant. The $Q$ factor can therefore only be improved through other means, such as by varying the ring properties.

# 6 Characterising the phonon interferometer

## 6.1 Quality factor measurements

We first explore the properties of the system with no applied rotation, $\Omega = 0$. This allows us to benchmark the performance of the interferometer, which will determine its maximum achievable sensitivity to rotation (see Sec. 7.5). We have performed experiments using the method described in Sec. 3.2 for a range of imprint amplitudes $0.0135 \lesssim \alpha_0 \lesssim 0.19$ and mode numbers $m^* \in \{3, 5, 7, 13\}$. An example series of density images for an $m^* = 7$ phase imprint is shown in Fig. 2(a–d), depicting the evolution of the oscillation over one period. The time evolution of the Fourier amplitude $A_{m^*}^{(I)}(t)$ obtained from Eq. (8) is shown in Fig. 2(e) (open circles), demonstrating the decaying oscillatory behaviour. From a fit to Eqs. (9)–(10) (solid

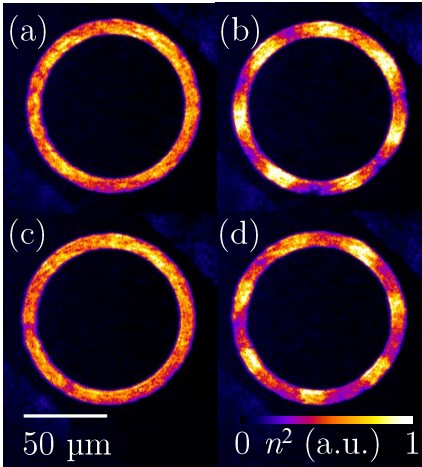
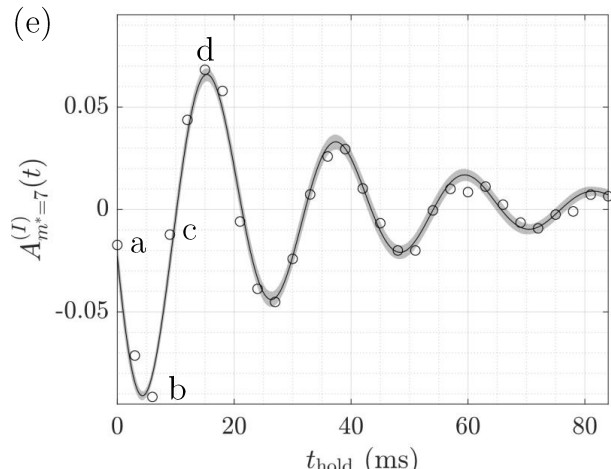

Figure 2: Typical experimental sequence for an $m^* = 7$ excitation with imprint time $t_{\text{imp}} = 400\,\mu\text{s}$ (imprint amplitude $\alpha_0 \approx 0.11$). (a)–(d) Density images for hold times of 0, 6, 9 and 15 ms respectively. The colour scale denotes the squared density in arbitrary units (a.u.). (e) Decaying oscillations of the imaginary component of the Fourier amplitude of the $m^* = 7$ azimuthal mode. Experimental data (open circles) have been fitted with a decaying sine wave (black line); the grey shaded region shows the 95% confidence interval for the fit. Data points corresponding to frames (a)–(d) are labelled accordingly.

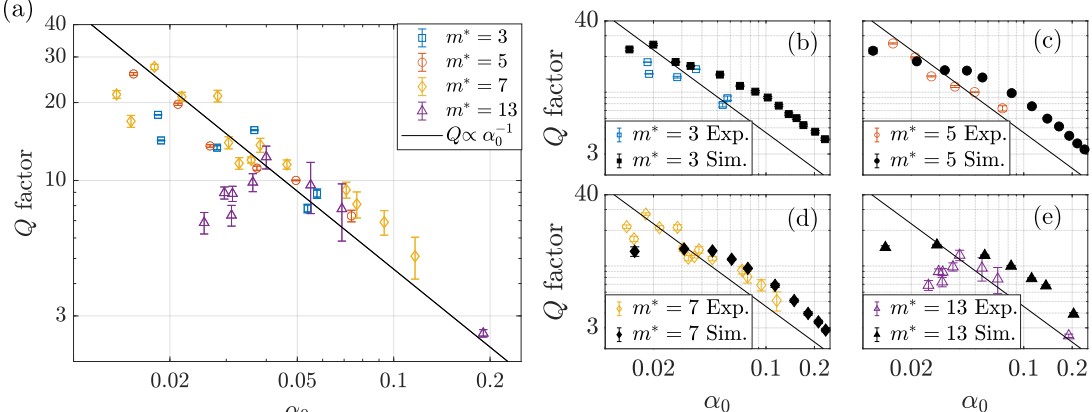

Figure 3: $Q$ factor as a function of imprint amplitude $\alpha_0$ for four different mode numbers $m^*$ (log–log scale). (a) Experimental data; (b)–(e) experimental (open coloured points) and c-field simulation (filled black points) data for $m^* = \{3, 5, 7, 13\}$, respectively. In all panels, the power-law fit to the experimental data, $Q \propto \alpha_0^{-1}$, is included for comparison. Standard errors in $Q$ are presented as vertical bars.

black line), we obtain measurements of the oscillation frequency $\omega_{m^*}$ and the decay time constant $\tau$, giving a quality factor $Q \approx 5$ for this configuration [Eq. (11)].

The $Q$ factors measured in this way for each run of the experiment are displayed in Fig. 3(a). In agreement with Ref. [1], we find $Q$ to be largely independent of the imprinted mode number $m^*$, meaning that performance cannot be improved through the choice of $m^*$ (see Sec. 5.2). Additionally, we find that the data approximately follow an inverse relationship with initial imprint amplitude, $Q \propto \alpha_0^{-1}$, indicating that we are in a nonlinear damping regime where $\tau$ is $\alpha_0$-dependent. In the limit $\alpha_0 \to 0$, we expect the system to evolve approximately linearly, corresponding to a constant $Q(\alpha_0)$. The $Q$ data do show a flattening at low $\alpha_0$ consistent with this expectation, although the spread of the data is too large to reveal a clear picture. From these results, we conclude that our interferometer should perform best in the limit of low $\alpha_0$. However, even the best performing configuration only achieves $Q \approx 27$—orders of magnitude lower than other modern rotation sensing technologies [5]. Moreover, using such weak imprints reduces the signal-to-noise ratio, until the oscillation becomes indistinguishable from noise at initial amplitudes $\alpha_0 \lesssim 0.013$.

To gain further insight into these results, we simulate our system using the c-field method outlined in Sec. 4 using a similar range of parameters $\alpha_0$ and $m^*$. We fit Eqs. (9)–(10) to the measured amplitude $A_{m^*}^{(I)}(t)$ of the imprinted mode over the first $\sim 100\,\mathrm{ms}$ of evolution. The resulting $Q$ measurements are shown as black points in Fig. 3(b)–(e), corresponding to $m^* = \{3, 5, 7, 13\}$, respectively. For comparison, the equivalent experimental data from (a) are also shown in each frame (b)–(e), along with the same power-law $Q \sim \alpha_0^{-1}$. We find broad agreement between our experimental and numerical results, to within the precision expected from the c-field methodology. For large $\alpha_0$ the numerical data follow the power-law quite convincingly. For small $\alpha_0$, the numerical results show a clear plateauing behaviour, particularly for $m^* = 7$ and $m^* = 13$, providing convincing evidence of linear decay behaviour in this regime. Importantly, the simulations do not predict significantly higher $Q$ factors for this experimental configuration, and hence the low $Q$ values observed cannot be attributed to imperfections in the experiment. Rather, the excited phonons are being damped by physical processes that are present in both simulation and experiment. In the next section, we attempt to understand these decay processes, as well as the observed transition between linear and nonlinear behaviour as $\alpha_0$ is varied.

## 6.2 Damping of phonons

### 6.2.1 Scattering processes

The imprinted excitation decays via mode mixing processes arising from the nonlinear interactions in the Bose gas. The dominant scattering processes involve interactions between three azimuthal quasiparticle excitations and the condensate mode.[3] In such a process, the imprinted mode $m^*$ will interact with two other quasiparticle modes (denoted by integers $p$ and $q$), as well as the condensate (denoted 0), leading to four potential events [24]: $m^*+p \to q+0$ (Landau damping [25,26]), $m^*+0 \to p+q$ (Beliaev damping [39,40]), $q+0 \to m^*+p$ (Landau growth) and $p+q \to m^*+0$ (Beliaev growth). In the absence of quantum fluctuations (which are neglected in the c-field simulations), these scattering events can only occur if both incoming modes and at least one of the outgoing modes have nonzero population. Additionally, the collisions must conserve both energy and angular momentum. Since we are interested in long wavelength excitations, the relevant part of the energy spectrum scales linearly with mode number, $E(m) \sim |m|$. In the case of Landau damping, the two conservation conditions are therefore $|m^*|+|p| = |q|$ (energy) and $m^*+p = q$ (angular momentum). Energy conservation prohibits interactions between counter-rotating modes, and hence we restrict our treatment to positive (anticlockwise) modes without loss of generality.

In the experiment, we expect Landau damping to dominate over all other processes because of the large initial population in the $m^*$ mode, as well as the large number of decay channels available for such collisions. (Beliaev damping, on the other hand, is highly restricted due to the low values of $m^*$, in combination with the conservation laws stated above.) We do, however, identify two unique Landau-type damping processes in the dynamics. Generally, the incoming mode $p$ will be thermally populated (i.e. $p \neq m^*$), leading to population transfer into another thermal mode $q = m^* + p$. However, in the special case $p = m^*$, a phenomenon known as frequency doubling occurs, where the outgoing mode $q = 2m^*$. We find that these two processes lead to significantly different damping behaviour of the imprinted mode, as explored in the next section.

### 6.2.2 Explanation of observed phonon damping

The two Landau-type decay processes described above become visibly distinct in the simulations for large imprints ($\alpha_0 \gtrsim 0.05$), manifesting as two separate exponential decay constants $\tau_1$ and $\tau_2$ in the decay of the envelope function $\alpha(t)$. This behaviour is demonstrated in Fig. 4 for an example c-field simulation with $m^* = 5$ and $\alpha_0 \approx 0.18$. In (a), the raw amplitude of the imprinted mode is plotted as a function of time (black circles), alongside two fits to Eqs. (9)–(10)—one for $t < 90\,\text{ms}$ (cyan solid line), and one for $t > 90\,\text{ms}$ (red dashed line). In (b), we plot the time-varying envelope $\alpha(t)$, which we have isolated by dividing by the sinusoidal component of the fit, $\sin(\omega_{m^*} t + \phi_0)$, with fitting parameters $\omega_{m^*}$ and $\phi_0$.[4] The exponential parts of the fit are also plotted for comparison, clearly demonstrating that two decay constants are required to describe the data. We have been unable to definitively observe this phenomenon in the experiment due to limitations in the signal-to-noise ratio, although the strongest imprint ($m^* = 13$, $\alpha_0 \approx 0.18$) shows behaviour consistent with two exponential decay rates.

Concurrent with this change in decay rate, we notice a change in the behaviour of the other azimuthal modes in the simulation. Immediately after $t = 0$, the higher harmonics of the $m^* = 5$ imprint ($m = 10, 15, ...$) begin to grow in population, as seen in Fig. 4(c). To quantify this effect, we have fitted quintic polynomial curves to approximate the envelopes $\alpha(t)$ of each

---

[3]Interactions between four quasiparticles are also possible, but are less frequent.

[4]We have implemented a moving-median binning procedure after dividing by the sinusoid in order to discount singularities that occur where it passes through zero.

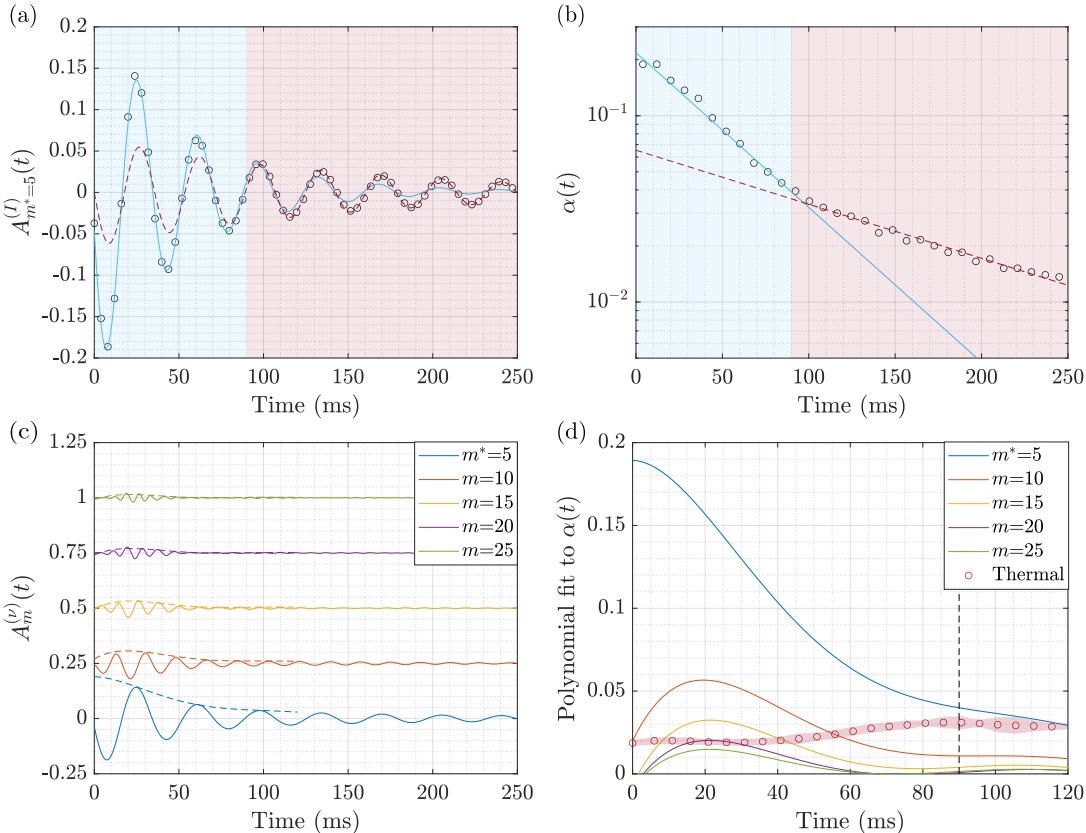

Figure 4: Exemplar of two decay rates in the c-field model for an imprint with $m^* = 5$ and $\alpha_0 \approx 0.18$, and an initial condensate fraction $n_0 \approx 0.81$. (a) Raw Fourier amplitude oscillation (black circles), and (b) the decay envelope function $\alpha(t)$ (black circles), obtained by dividing the data by $\sin(\omega_{m^*} t + \phi_0)$, with fitted frequency $\omega_{m^*} = 2\pi \times 28$ Hz and phase offset $\phi_0 = 3.3$ rad (blue shaded region) and $\phi = 3.1$ rad (red shaded region). Note the semi-log scale in (b). In both (a) and (b), the envelope function is well described by Eq. (10), with time constants $\tau_1 \approx 50$ ms for $t < 90$ ms (blue shaded region), and $\tau_2 \approx 143$ ms for $t > 90$ ms (red shaded region). The fits are shown with a cyan solid line and a red dashed line, respectively. (c) Amplitudes of the imprinted mode $m^* = 5$ and higher harmonics (solid lines), vertically offset for clarity. Dashed lines show fifth-order polynomial fits capturing the envelope $\alpha(t)$ of the amplitude data. The index $\nu \in \{R, I\}$, with real component plotted for even modes, and imaginary component plotted for odd modes. (d) Comparison of fitted amplitude envelope polynomials (solid lines), as well as the thermal mode population $\Lambda_{\text{th}}(t)$ (see text). The red circles correspond to time-averaged values of $\Lambda_{\text{th}}$, while the shaded region denotes the standard deviation. The vertical dashed line denotes the 90 ms timescale where the decay rate crosses over, as measured from (a) and (b).

of these modes. The fits are plotted in Fig. 4(d) [also shown as dashed lines in (c)]. The higher harmonics are seen to rapidly grow in amplitude at early times, before peaking at $\approx 20$ ms with a maximum population that decreases for increasing $m$. This growth of amplitude can be explained as a form of high harmonic generation (HHG) [41,42] arising from a cascade of Landau-type processes $im^* + jm^* \rightarrow (i + j)m^* + 0$. Initially, the frequency doubling process described above ($i = j = 1$) converts population into the $2m^*$ mode. This then stimulates

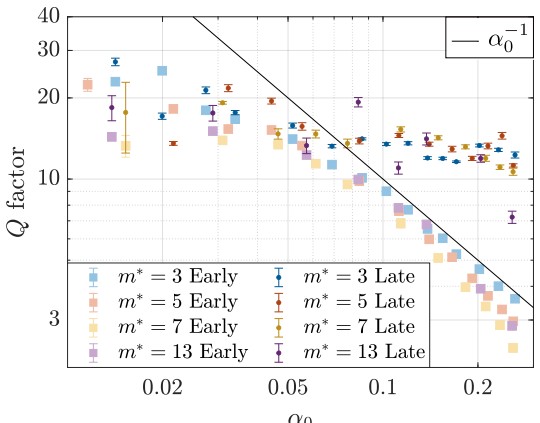

Figure 5: $Q$ factor as a function of the initial imprint amplitude $\alpha_0$ for different imprinted mode numbers $m^*$ from the c-field simulations with $n_0 \approx 0.81$. For $\alpha_0 \gtrsim 0.05$, two decay times $\tau_1$ and $\tau_2$ emerge, giving rise to two distinct $Q$ values, depending on the temporal window used for the fit ('early' or 'late', respectively). We attribute the source of the $Q \sim \alpha_0^{-1}$ power-law to be the HHG damping process, while the approximately $\alpha_0$-independent behaviour reflects the Landau scattering from thermal modes. The 'early' data are the same as those plotted in Fig. 3(b–e).

$(i = 1, j = 2)$ collisions into the $3m^*$ mode, followed by growth of the $4m^*$ mode via $(i = 1, j = 3)$ and $(i = 2, j = 2)$, and so on. In this way, the population rapidly transfers from the imprinted mode to the higher harmonics. Other modes that are not integer multiples of $m^*$ are initially thermally populated, and their amplitudes remain at $|A_m(t)| \lesssim 0.01$ throughout the simulation.

After peaking at $\approx 20\,\text{ms}$, the populations of the higher harmonics begin to decay again, facilitated by scattering processes involving the thermally populated modes. Eventually, they decrease to the level of the thermal modes, indicating that the HHG process is no longer dominant. Importantly, around the observed crossover time of $\approx 90\,\text{ms}$, the population in the imprinted mode becomes almost equal to the population across all the thermal modes, $\Lambda_{\text{th}}(t) \equiv \sum_{m \neq jm^*} |A_m(t)|$, as seen in Fig. 4(d).[5] Once this occurs, the thermal Landau damping process should begin to dominate for mode $m^*$, because collisions with thermal quasiparticles at least as probable as collisions with imprinted quasiparticles.

With this understanding, we revisit the numerical results presented in Fig. 3(b–e). By performing additional fits of Eqs. (9)–(10) to $A_{m^*}^{(I)}(t)$ using a later temporal window, we find that a second decay time $\tau_2$ systematically emerges at large $\alpha_0$. Figure 5 shows the $Q$ factor as measured for this later fitting window (dark circles), with the data from the earlier fitting window (light squares) also shown for comparison. Two distinct values of $Q$ are obtained for $\alpha_0 \gtrsim 0.05$. While the early time data follow $Q \sim \alpha_0^{-1}$ in this region (as identified in Sec. 6.1), the late time data show little dependence on $\alpha_0$. This suggests that while the early HHG damping process gives rise to nonlinear decay, once the late time thermal Landau damping process takes over, the dynamics enter an approximately linear regime. The weak downward trend of $Q(\alpha_0)$ from the late fits is also consistent with an increase in heating resulting from the redistribution of increasingly large imprinted quasiparticle populations into the thermal modes.

---

[5]Note that the thermal mode amplitudes oscillate in time, meaning that this measure will on average underestimate the true thermal population.

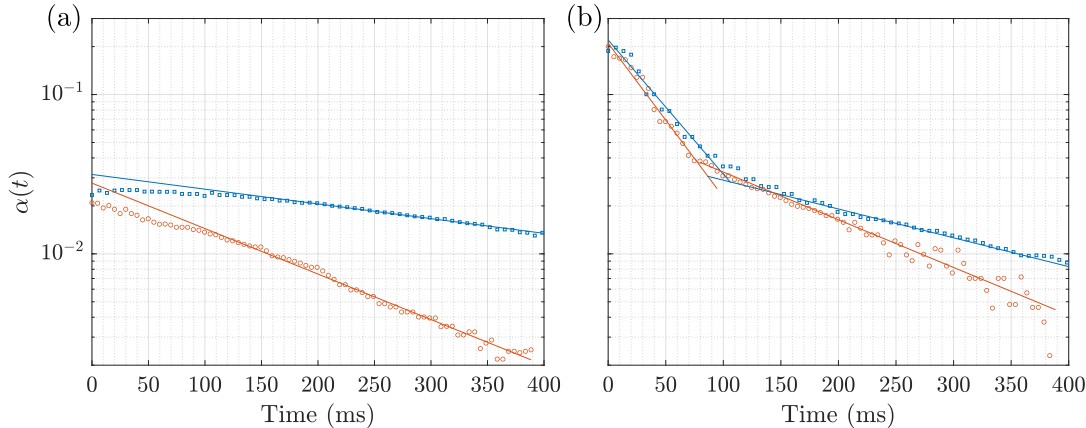

Figure 6: Envelope functions $\alpha(t)$ from the c-field simulations for an $m^* = 5$ imprint at condensate fractions $n_0 = 1$ (blue squares) and $n_0 \approx 0.81$ (orange circles), obtained by dividing each amplitude curve $A_{m^*}^{(I)}(t)$ by the sinusoidal component of the fit to Eqs. (9)–(10). (a) Low initial amplitude imprint, $\alpha_0 \approx 0.02$; (b) high initial amplitude imprint, $\alpha_0 \approx 0.25$. Solid lines show exponential fits to Eq. (10). In (b), the early-time fits are performed in the window $t \leq 75\,\mathrm{ms}$; all other fits correspond to windows $t \geq 150\,\mathrm{ms}$.

### 6.2.3 Temperature dependence of damping processes

Due to their different origins, the HHG and thermal Landau damping processes should give rise to distinct behaviour as both the imprint amplitude and the temperature of the system are varied. For large $\alpha_0$ where HHG initially dominates, lowering the temperature should not significantly affect the decay rate, since the frequency doubling process is unaffected by the thermal populations. By contrast, small $\alpha_0$ imprints should be strongly temperature dependent, since thermal Landau damping is the dominant process. In the limit of both zero temperature and small $\alpha_0$, damping should be minimised, because only very weak HHG should be possible.

To test this hypothesis, we have performed additional simulations of the $m^* = 5$ imprint at zero temperature (*i.e.* condensate fraction $n_0 = 1$). Figure 6 compares the envelope functions $\alpha(t)$ for $n_0 = 1$ (blue squares) and $n_0 \approx 0.81$ (orange circles) for two imprint amplitudes $\alpha_0$. The envelopes have been obtained in the same way as in Fig. 4(b). For weak imprints [panel (a)], we find that the decay rate is much higher at finite temperature than zero temperature. Consistent with our prediction, the envelope function for $n_0 = 1$ is approximately flat at early times, showing a gradual approach to exponential decay as the population transfers to higher harmonics of $m^*$. At the highest amplitudes [panel (b)], we see that the early-time decay rate is similar at the two temperatures, demonstrating the temperature-independence of HHG. The late-time decay rate, on the other hand, is significantly higher at finite temperature, consistent with the prediction that thermal Landau processes are more important here.

### 6.2.4 Change in quality factor with temperature

The data in Fig. 6(a) indicate that it should be possible to improve the $Q$ factor of our system by reducing the temperature. To explore this possibility further, we have repeated our simulations of the $m^* = 5$, $\alpha_0 \approx 0.02$ imprint for a range of condensate fractions $n_0$. As seen in Fig. 6(a), the envelope $\alpha(t)$ exhibits a slow approach to exponential decay at the lowest temperatures. We therefore measure the decay constant $\tau$ using two temporal windows: $t \leq 100\,\mathrm{ms}$ and $150\,\mathrm{ms} \leq t \leq 400\,\mathrm{ms}$. The resulting $Q$ factors are presented in Fig. 7. Evidently, the early time

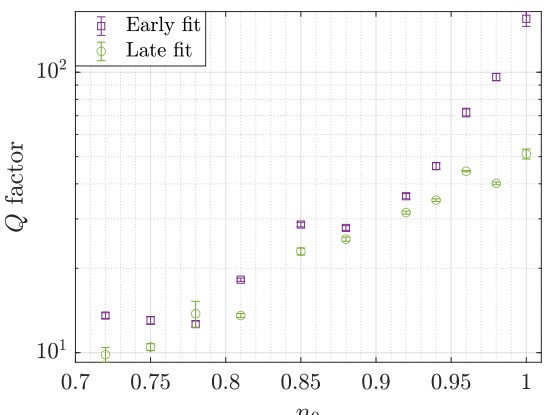

Figure 7: $Q$ factor as a function of condensate fraction $n_0$ in the c-field model, for an $m^* = 5$, $\alpha_0 \approx 0.02$ imprint. 'Early' and 'late' fits correspond to temporal windows $t \leq 100\,\mathrm{ms}$ and $150\,\mathrm{ms} \leq t \leq 400\,\mathrm{ms}$, respectively. Standard errors are shown.

$Q$ factors are significantly larger at the highest condensate fractions, $n_0 \gtrsim 0.95$. In this limit, $Q$ factors as large as $\sim 150$ are attainable, provided that the measurement only probes the first $\sim 100\,\mathrm{ms}$ of evolution. If longer interrogation times are required, the maximum possible $Q$ drops to $\sim 50$, which is only a minor improvement over our best experimentally measured values.

## 7 Rotation measurement with a persistent current

### 7.1 Persistent currents

We now turn to testing the performance of our superfluid ring as a rotation sensor in the case rotation is applied, *i.e.* $|\Omega| > 0$. Due to the challenges involved with rotating the laboratory FoR, we have instead opted to induce rotation in the ring by generating persistent currents. This provides a proof-of-principle demonstration of rotation sensing using phonon interferometry [14]. However, we note that in a realistic rotation sensing application, persistent currents are undesirable. Such currents will be generated if the atoms are rotating faster than the lowest rotation rate the ring can support, $\Omega_0$, when the BEC is formed, introducing ambiguity into the rotation measurement.

A superfluid confined to a ring geometry is capable of supporting persistent currents, which arise due to the quantisation of circulation around the ring in units of $h/m_{\mathrm{a}}$, where $h$ is Planck's constant. The rotation rate $\Omega$ is therefore also quantised such that:

$$\Omega = l\Omega_0 \approx \frac{l\hbar}{m_{\mathrm{a}}R^2}, \tag{12}$$

where $l$ is number of circulation quanta and $R$ the radius of the ring. With a ring radius of $45\,\mu\mathrm{m} \lesssim R \lesssim 55\,\mu\mathrm{m}$, the presence of a single quantum of circulation (*i.e.* $l = 1$) indicates a rotation rate that varies within the range $0.24\,\mathrm{rad/s} \lesssim \Omega_0 \lesssim 0.36\,\mathrm{rad\,s^{-1}}$ as a function of radius.

### 7.2 Experimental stirring protocol

We generate persistent currents experimentally by stirring with a potential barrier [14], as depicted in Fig. 8. First, we load the atoms into a ring-shaped trap with inner and outer radii

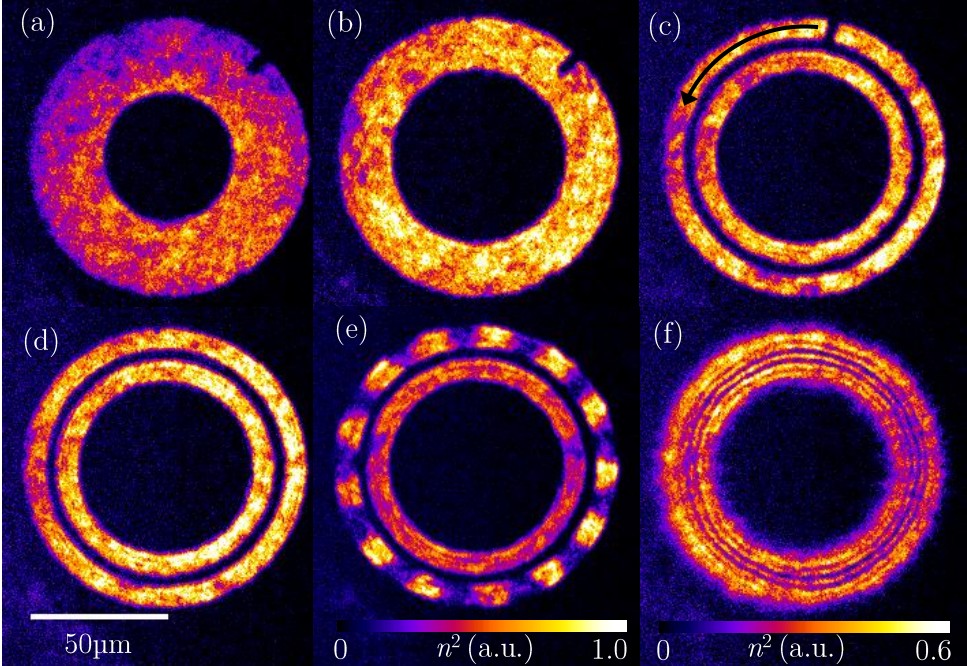

Figure 8: Absorption images of the squared density during the stirring sequence. (a) A thick ring is created, with a barrier for azimuthal stirring (top right); (b) the inner radius is expanded to increase the atomic density; (c) a radial barrier is introduced and the azimuthal barrier is rotated to generate a winding in the outer ring; (d) the azimuthal barrier is removed; (e) either a sinusoidal phase is imprinted in the outer ring (in this case, $m^* = 13$), or; (f) the inner and outer ring are allowed to interfere for a direct measurement of the winding number (in this case, $l = 1$). The squared density is measured in the same arbitrary units (a.u.) across all frames. The color scale in (f) applies to frames (a–d,f), while the range of the color scale for frame (e) has been increased for clarity.

$r_{\text{in}} \approx 25\,\mu\text{m}$ and $r_{\text{out}} \approx 55\,\mu\text{m}$, including an additional barrier to be used for azimuthal stirring [panel (a)]. We then expand the inner radius to $r_{\text{in}} \approx 30\,\mu\text{m}$ in order to increase the atomic density [panel (b)]. A radial barrier of width $\approx 10\,\mu\text{m}$ is then ramped on, splitting the system into two concentric rings of $\approx 10\,\mu\text{m}$ thickness [panel (c)]. The barrier is then swept around the outer ring for $< 1$ cycle, moving with an angular acceleration of $1.53\,\text{rad}\,\text{s}^{-2}$ until a desired angular velocity is achieved in the outer ring. At this point, we remove the azimuthal barrier and allow the system to relax for $500\,\text{ms}$ so that any excitations generated by the stirring can dissipate [panel (d)].

To use the system as a rotation sensor, we then imprint phonons into the outer rotating ring in the same way as described in Sec. 3.2, ignoring the inner ring [panel (e)]. We can then proceed to measure the rotation rate of the imprinted phonon mode by measuring the rate of change of the angle $\theta(t) = \arg\{A_{m^*}(t)\}$ (see Sec. 5). Alternatively, we can perform a 'control' measurement of the system's rotation rate by removing the trapping potential and letting the two rings interfere before taking an image [panel (f)]. The resulting interference pattern is expected to contain $l$ fringes, enabling a direct measurement of the circulation [43, 44]. By comparing these two measurements, we can confirm that the phonon interferometry scheme matches an established method of measuring rotation.

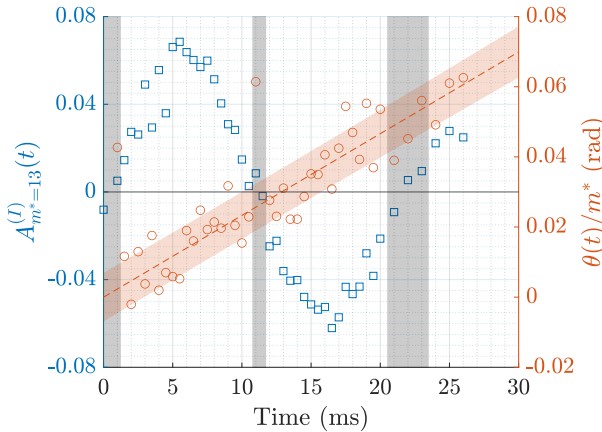

Figure 9: Evolution of the Fourier amplitude for an $m^* = 13$ imprint, with a persistent current of $\approx 2.5\,\mathrm{rad\,s^{-1}}$ around the ring. Blue squares show the imaginary component of $A_{m^*}(t)$ (left axis), while orange circles show the phase $\theta(t)/m^*$ (right axis). The orange dashed line shows a linear fit to the phase, with the shaded region denoting one standard error. Vertical grey shading corresponds to regions with small imaginary component ($|A_{m^*}^{(I)}(t)| \leq 0.01$) where phase data are excluded from the fit.

## 7.3 Experimental rotation measurement

We perform rotation measurements for a range of circulation quanta around the ring, up to $l = 9$ (corresponding to $\Omega \approx 2.5\,\mathrm{rad\,s^{-1}}$). For these measurements, we use phonon modes $m^* = 7$ and $m^* = 13$ to maximise the signal-to-noise ratio in the measurement of the phase accumulation rate $\mathrm{d}\theta/\mathrm{d}t = m^*\Omega$. Figure 9 displays an example measurement for an $m^* = 13$, $l = 9$ imprint, with both the imaginary component of the Fourier amplitude $A_{m^*}^{(I)}(t)$ and the angle $\theta(t)$ shown. The dashed orange line shows a linear fit to $\theta(t)$, from which the rotation rate $\Omega$ can be measured. Points where the imprinted mode amplitude is small ($|A_{m^*}^{(I)}(t)| \leq 0.01$, vertical shaded regions) are excluded from the fit, as these correspond to times when the density profile is approximately uniform around the ring, resulting in an undefined angle $\theta$.

Figure 10(a) displays the rotation rate $\Omega_{\mathrm{phonon}}$ as measured from phonon interferometry, compared to $\Omega_{\mathrm{fringe}}$ obtained from fringe counting (see Sec. 7.2). All points on the graph are located close to the ideal scenario $\Omega_{\mathrm{phonon}} = \Omega_{\mathrm{fringe}}$ (solid line), demonstrating a successful implementation of phonon interferometry for rotation sensing. However, our stirring sequence does not provide deterministic control of the winding number, introducing variability in the applied rotation rate. This may explain the large spread in $\theta(t)$ seen in Fig. 9, which is in turn reflected in the vertical error bars in Fig. 10(a). The horizontal error bars are dominated by the variation in rotation frequency over the finite width of the ring due to the irrotational nature of the flow (see Sec. 7.1). We note that both of these effects are artefacts of the use of persistent currents for rotation. In a realistic rotation sensing scenario, the BEC would rotate as a rigid body relative to the laboratory FoR, and these sources of uncertainty would not be present. We discuss the fundamental rotation sensitivity of this system in Sec. 7.5.

## 7.4 Numerical rotation measurement

We simulate the rotation measurements using the c-field model for comparison with the experimental results. We use the same simulation procedure outlined in Sec. 4, with an additional step of phase imprinting $l$ circulation quanta, $\psi \to \psi e^{il\theta}$, onto the ground state before evolving Eq. (5) to equilibrium at finite temperature. We perform simulations with $l = \{1, 2, 3, 6, 9\}$ and imprinted mode numbers $m^* = \{7, 13\}$.

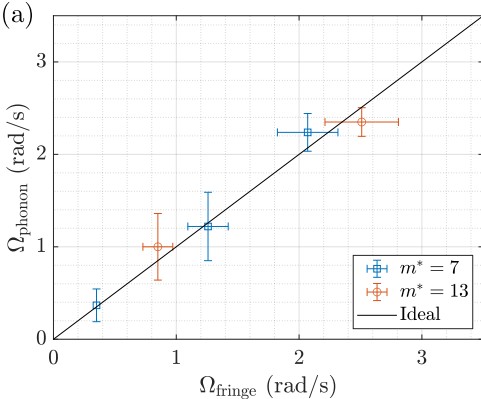
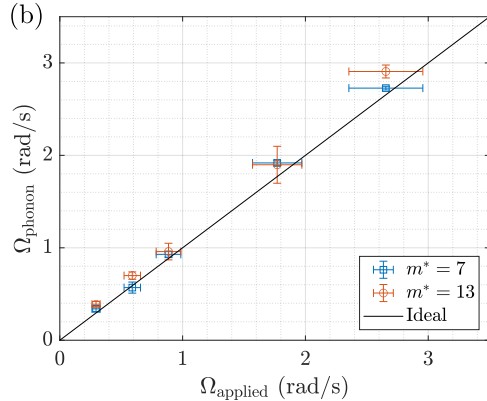

Figure 10: Measured rotation rates $\Omega_{\text{phonon}}$ from phonon interferometry with $m^* = 7$ and $m^* = 13$ imprints, as a function of applied rotation. (a) Experimentally measured rotation rates as a function of the rotation as measured from the fringe count $\Omega_{\text{fringe}}$; (b) rotation rates measured from the c-field simulations as a function of the applied rotation $\Omega_{\text{applied}}$. In both frames, the ideal result is shown as a black line. Note that the winding number $l$ is nondeterministic in the experiment, whereas the simulations provide precise control over $l$. The imprint amplitude $0.04 \lesssim \alpha_0 \lesssim 0.08$ for the experimental data, while $\alpha_0 \approx 0.1$ was chosen for the simulations. The error bars indicate the standard error.

Following the same analysis procedure as the experiment, we extract the rotation rate $\Omega_{\text{phonon}}$ by measuring the rate of change of $\theta(t)$. As for the experimental data, we remove points where $|A_{m^*}(t)| \leq 0.01$ in order to avoid singularities, and we restrict the fits to $t \lesssim 100$ ms. We then bootstrap our fit by repeatedly sampling 10 randomly selected points out of the remaining $\sim 30$ points in the time sequence, performing a linear fit to each subsample. The rotation rate and its uncertainty are then obtained from the mean and standard deviation of the resulting distribution of gradients. These measurements are plotted as a function of the applied rotation $\Omega_{\text{applied}} = l\Omega_0$ in Fig. 10(b). The data follow the expected behaviour, although the singularities do still affect the quality of the fitting, leading to some slight disagreement with the expected trend. Evidently, the simulation results are more precise than those from the experiment, with smaller vertical error bars in particular due to the absence of uncertainty in the applied winding number. However, the horizontal error bars remain approximately the same size as the experimental data, due to the aforementioned variation in rotation frequency over the width of the ring. This result demonstrates the viability of the sensor for practical rotation measurement and confirms our understanding of the experiment.

## 7.5 Sensitivity of rotation sensor

The performance of the interferometer as a rotation sensor can be quantified by its sensitivity $\Delta\Omega$, defined as the smallest rotation frequency it is capable of resolving. Unlike in the previous sections, here we focus on a scenario where just a single run of the experiment is performed, and a sample is obtained at an optimal time $t$. The uncertainty in a measurement of $\Omega$ made from this sample is $\Delta\Omega(t) = \Delta A_{m^*}|\mathrm{d}A_{m^*}/\mathrm{d}\Omega|^{-1}$, which from Eqs. (9)–(10) gives

$$\Delta\Omega(t) = \Delta A_{m^*} \frac{\exp(t/\tau)}{m^*\alpha_0 t |\sin(\omega_{m^*} t + \phi_0)|}. \tag{13}$$

If we make a measurement at time $t = \tau$, the ratio $\exp(t/\tau)/t$ will be minimised, and $\Delta\Omega(\tau) \propto 1/\tau|\sin(\omega_{m^*}\tau + \phi_0)|$. The global minimum of the single-shot sensitivity will therefore be achieved if $t = \tau$ also corresponds to a maximum of the sinusoid (*i.e.* a maximum of the density perturbation), giving:

$$\Delta\Omega_{\mathrm{min}} = \Delta A_{m^*}\frac{\mathrm{e}}{m^*\alpha_0\tau}\,. \tag{14}$$

This optimal scenario could be engineered via a change in the frequency $\omega_{m^*}$, for example by adjusting the atomic density.

Expressing this ideal sensitivity in terms of the $Q$ factor (11),

$$\Delta\Omega_{\mathrm{min}} \approx \Delta A_{m^*}\frac{\mathrm{e}\omega_0}{2\alpha_0 Q}\,, \tag{15}$$

it becomes evident that the imprinted mode number $m^*$ does not affect the sensitivity, and that $\Delta\Omega_{\mathrm{min}}$ will improve with larger $Q$. However, in Sec. 6.1 we found that $Q$ and $\alpha_0$ are not independent quantities in general, and in particular that $Q \propto \alpha_0^{-1}$ for strong imprints. In this regime, the sensitivity becomes completely independent of the properties of the imprinted mode, leaving only its dependence on the system properties via $\omega_0$.

If the density profile measurement is shot-noise limited, we expect fluctuations in the Fourier amplitude of order $\Delta A_{m^*} = (2N_{\mathrm{atoms}})^{-1/2}$ [1]. In this limit, Eq. (15) predicts an optimal sensitivity of $\Delta\Omega_{\mathrm{ASN}} \approx 0.04\,\mathrm{rad\,s}^{-1}$ for our experiment. For comparison, we directly estimate the fluctuations in $A_{m^*}(t)$ by experimentally measuring the spread $\Delta A_{m^*}^{(R)}(t)$ for zero imprinted rotation, in which case $\langle A_{m^*}^{(R)}(t)\rangle = 0$ is expected due to the orientation of the phase imprint. By doing this for all non-rotating datasets presented in Fig. 3(a), we estimate that $\Delta A_{m^*}$ is approximately eight times larger than than the shot-noise limited value, leading to an achievable sensitivity of $\Delta\Omega_{\mathrm{min}} \approx 0.3\,\mathrm{rad\,s}^{-1}$ from a single measurement at $t = \tau$.

# 8 Conclusions

We have established the sensitivity of a rotation sensing schemed based on the interference of counter-propagating phonon modes imprinted in the phase of a ring-shaped Bose–Einstein condensate. Such a device is fundamentally limited by the lifetime of the imprinted excitations, and the system achieves quality factors up to a maximum of $Q \approx 27$. This is an improvement over previously reported values in similar experimental setups [1,37], but still many orders of magnitude poorer than commercially available sensors.

By performing c-field simulations of the experiment, we have confirmed that the low quality factors obtained do not arise from experimental deficiencies. Rather, the decay of the imprinted excitations is caused by physical damping processes facilitated by the nonlinearity of the BEC. Intriguingly, the numerics have revealed two distinct decay processes at work. Scattering between imprinted and thermal quasiparticles provides a form of damping that depends only on temperature, while collisions between imprinted quasiparticles give rise to damping that depends on the imprinting amplitude but not on temperature. In the limit of zero temperature and weak imprints, where both of these damping mechanisms should be minimised, our simulations suggest that $Q$ factors up to $\sim 150$ may be attainable in this system.

We have successfully performed a rotation measurement using this phonon interferometry scheme by creating a persistent current in the ring BEC, thereby demonstrating a resolution below the fundamental frequency of the ring, $\Omega_0 \approx 0.29\,\mathrm{rad\,s}^{-1}$, when fitting to a sequence of measurements. We predict that the optimal rotation sensitivity attainable from a single run of our experiment is $\Delta\Omega_{\mathrm{min}} \approx 0.3\,\mathrm{rad\,s}^{-1}$. While this is an improvement over previous work [1], it is significantly above the atomic shot-noise limit of $\Delta\Omega_{\mathrm{ASN}} \approx 0.04\,\mathrm{rad\,s}^{-1}$ of our apparatus.

The rotation sensing precision could therefore be improved by reducing the noise floor, for example via the reduction of optical noise sources. Improving the signal-to-noise ratio of the measurement would also allow for weaker imprints to be used, increasing the sensitivity [see Eq. (15)]. Further gains could be made in the sensitivity by implementing non-destructive imaging, which would allow for multiple sequential measurements without increasing the experimental run time [45]. Finally, it might be possible to alter the trap geometry to limit the available scattering channels, making the imprinted excitations resistant to decay.

Significant improvements in sensitivity would be required to produce a sensor of comparable precision to those already commercially available. However, the superfluid's absolute FoR does prevent measurement drift, potentially providing a very significant potential advantage for operation over extended periods. These devices may therefore prove useful if used in conjunction with conventional rotation sensors as a means of calibration to reduce uncertainty in classical systems over prolonged operation [46–48]. This could be achieved by repeatedly performing single-shot rotation rate measurements using the phonon system. The difference between these measurements and those from the classical system from individual measurements would accumulate to provide a long term drift value that could thus be subtracted from the classical measurement [46–48].

# Acknowledgements

**Funding information**   Funding for this work was provided by a Next Generation Technologies Fund – Quantum Technologies grant (QT95) by the Commonwealth Defence Science and Technology Group, Australian Recearch Council Centre of Excellence for Engineered Quantum Systems (EQUS, CE170100009), and ARC Discovery Projects grant DP160102085. This research was also partially supported by the Australian Research Council Centre of Excellence in Future Low-Energy Electronics Technologies (project number CE170100039). T.W.N. and S.A.H. acknowledge the support of Australian Research Council Future Fellowships FT190100306 and FT210100809, respectively.

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
