# Peer review of "Viability of rotation sensing using phonon interferometry in Bose-Einstein condensates"

_SciPost Physics, doi:SciPost Phys. 15, 128 (2023)_

## Round 1 · Referee Report · Anonymous (Referee 1) · 2023-1-19

Strengths

  1. Very interesting paper on phonon interferometry in BEC;
  2. Significant improvement of sensitivity of a rotation sensor;
  3. Good combination of experiment and theory;
  4. Nice theoretical discussion of phonon decay mechanisms;
  5. Well written.

Weaknesses

Limited discussion about the control of temperature in the experimental protocol.

Report

This paper is about phonon interferometry in a ring-shaped Bose-Einstein condensate. Building on similar ideas developed in previous papers (refs. [1] and [14], for example), the authors introduce a new experimental protocol in which they create and observe density patterns made of counter-propagating phonons. If the entire ring rotates, such a pattern should also rotate. As a test, the authors induce persistent currents in the ring and then observe the consequent rotation of the phonon pattern finding good agreement with expectations. The analysis of the experiment is nicely integrated with numerical simulations with a c-field approach (namely, a stochastic projected Gross–Pitaevskii equation). The theory allows them to confirm the validity of the protocol, adding relevant information about the main decay mechanisms and the role of the key parameters, and also suggesting ways to further improve the sensitivity of their rotation sensor.

The paper is well written. The tools used both for the experiments and the theory are appropriate. The results are original and interesting. I recommend publication.

These below are minor comments, which could be used to improve the presentation.

i) In Fig.1, the red line is very thin and scarcely visible.

ii) The transverse trapping frequency is 133 Hz. With this value, is the density profile along $z$ well approximated by a TF inverted parabola? If yes, what is the TF radius compared to the width the ring? Related to this questions, I suggest the authors to explicitly mention that the thermodynamic of this system is 3D (BEC) and not 2D (BKT), while the relevant dynamics is 1D (i.e., along the ring only).

iii) It seems that only one value of temperature is used in the experiments, such as the condensate fraction is about 0.8. Nothing is said about the experimental control of $T$. Can one vary $T$, for instance going to lower $T$ in order to increase the quality factor? Is the experimental $T$ estimated only by comparing the phonon decay time with numerical simulations? Are there ways to independently estimate the ratio $T/T_c$, where $T_c$ is the BEC transition temperature in the ring?

iv) When introducing the theoretical model (c-field), the authors write: "We emphasise that this model is not expected to provide quantitative predictions for the dynamics of the Bose gas, and should be treated as indicative". I think this is too pessimistic. There are cases where the stochastic GPE is able to quantitatively reproduce the dynamics of condensates at finite temperature. An example, which is not very dissimilar to the present one, except for the rotations, is given in PRL 121, 145302 (2018). Even in the present case, it seems that the theory is not purely qualitative.

v) After eq.(5), the authors say that the c-field is a "set of plane waves". They should clarify if $\bf k$ is a vector in 3D, and not only in the $xy$-plane. If yes, this means that they use a plane wave basis also along $z$, where the confinement is harmonic. What about using a more realistic harmonic oscillator basis along $z$? Would they expect any difference in the simulations?

vi) Toward the end of section 4, the authors say that the above-cutoff atoms have been neglected in the calculation of the condensate fraction. Do they have any crude estimate of the effect of such an approximation? Related to this question, I see that they choose a cutoff energy of the same order of the chemical potential. Did they try to check what happens by changing it?

vii) Soon after they say "we find that 2D simulations provide quantitatively similar predictions for the decay of the imprinted excitation", but this sentence is somewhat cryptic. In a purely 2D gas in the same ring, the transition would be of BKT type and the condensate fraction would not behave in the same way. So, when comparing 2D with 3D, what is kept fixed? Condensate fraction? $T/T_c$? Number of atoms? As it is, this comment is not useful and might be a source of confusion.

viii) In the caption of fig.8, I guess $l=9$, not $l=1$. Also, I think it is better to write the values of both $m^*$ and $l$.
  • validity: high
  • significance: high
  • originality: good
  • clarity: high
  • formatting: excellent
  • grammar: perfect

Author:  Charles Woffinden  on 2023-06-12  [id 3724]

(in reply to Report 1 on 2023-01-19)
Category:
answer to question
validation or rederivation

We thank the referee for their constructive feedback. We believe that the manuscript is now stronger as a result of the changes made to address these comments. Please find below our response to each point, including descriptions of any changes made to the manuscript.

THE REFEREE WRITES:
i) In Fig.1, the red line is very thin and scarcely visible.

WE RESPOND:
We thank the referee for pointing this out. We have made the line thicker.

THE REFEREE WRITES:
ii) The transverse trapping frequency is 133 Hz. With this value, is the density profile along z well approximated by a TF inverted parabola? If yes, what is the TF radius compared to the width the ring? Related to this questions, I suggest the authors to explicitly mention that the thermodynamic of this system is 3D (BEC) and not 2D (BKT), while the relevant dynamics is 1D (i.e., along the ring only).

WE RESPOND:
Yes, we expect that the density profile in all three dimensions should be well approximated by a Thomas--Fermi profile. The density profile in $z$ is approximated to be an inverted parabola, with a Thomas--Fermi radius of $\approx 4\mu m$ (diameter of $\approx 8\mu m$), which is on the same order as the 2D annulus width of $\approx 10\mu m$. Based on this, the thermodynamics of the system are indeed expected to be 3D. We have updated Sec. 3.1 to make this point.

THE REFEREE WRITES:
iii) It seems that only one value of temperature is used in the experiments, such as the condensate fraction is about 0.8. Nothing is said about the experimental control of T. Can one vary T, for instance going to lower T in order to increase the quality factor? Is the experimental T estimated only by comparing the phonon decay time with numerical simulations? Are there ways to independently estimate the ratio T/T$\mathrm{\mathbf{c}}$, where T$\mathrm{\mathbf{c}}$ is the BEC transition temperature in the ring?}

WE RESPOND:
A condensate fraction of $\sim 0.8$ corresponds to the lowest temperature we were able to achieve in the experiment. We have determined this fraction by performing a bimodal fit on an image of the condensate density at a particular time of flight. The two fits used are a Gaussian (approximating a Boltzmann distribution) for the thermal component and a Thomas-Fermi distribution for the condensate fraction. The proportion of atoms contained within the Thomas-Fermi distribution is the condensate fraction. While the experimental temperature isn't explicitly stated in the paper, this can be calculated by acquiring multiple images at different times of flight and performing the same fitting routine. The expansion of the thermal cloud over time, and hence the waist of the Gaussian profile, informs us of the temperature of the thermal fraction.
One could evaporate less and thus leave a lower condensate fraction; however, we opted not to do this in the experiment, instead focusing on minimising the temperature so that the quality of the resonator is maximised. We have now updated the text in Section 3.1 to address this.

THE REFEREE WRITES:
iv) When introducing the theoretical model (c-field), the authors write: "We emphasise that this model is not expected to provide quantitative predictions for the dynamics of the Bose gas, and should be treated as indicative". I think this is too pessimistic. There are cases where the stochastic GPE is able to quantitatively reproduce the dynamics of condensates at finite temperature. An example, which is not very dissimilar to the present one, except for the rotations, is given in PRL 121, 145302 (2018). Even in the present case, it seems that the theory is not purely qualitative.

WE RESPOND:
Although we are enthusiastic users of c-field methods for simulating finite temperature Bose-Einstein condensates, we do want to emphasize that care must be taken. While it is true that it can sometimes quantitatively reproduce dynamics, sometimes fitting parameters are necessary, such as the exact value of the cutoff. The paper that the referee refers to is on the decay of collisionless sound, and it makes the following comment:
"In RPA, the behavior of $Q$ is the consequence of Landau damping, i.e., the coupling between the collective sound oscillation and the (thermally populated) single particle excited states included in the ideal Bose gas response (2) (see also Ref. [39] for similar results). In SGPE, the same mechanism is accounted for by the dynamical coupling between excited states described by the classical field below the cutoff energy. This is confirmed by the independence of Q on frequency, as shown in the inset of Fig. 4; in fact, if damping were collisional, it would exhibit a quadratic increase with $\omega$ and hence a pronounced frequency dependence of the quality factor."

So it seems reasonable that in this work that the dynamics are relatively independent of the cutoff. However, in our simulations the damping is purely due to collisions - and so we feel that more care must be taken. C-field methods do not capture the behaviour of high-energy modes correctly (the momentum occupation follows a classical equipartition distribution, rather than a Bose-Einstein one). So, in general they cannot quantitatively predict all relevant observables of the Bose gas simultaneously, as explored in e.g. Phys. Rev. A 92, 063620 (2015). We therefore think it should be made clear to the reader that this model is not fully quantitative.

Regardless, we have revised the text to be less pessimistic, as the referee suggests:
"Although c-field models cannot quantitatively capture all aspects of the Bose gas simultaneously~[28], they have nonetheless been shown to match experiments when used appropriately~[29-31]. Here we use the c-field simulations to provide insight into the mechanisms of the damping behaviour of the system at non-zero temperature, and find that the results are in semi-quantitative agreement with experiments when the condensate fractions are matched."

THE REFEREE WRITES:
v) After eq.(5), the authors say that the c-field is a "set of plane waves". They should clarify if k is a vector in 3D, and not only in the xy-plane. If yes, this means that they use a plane wave basis also along z, where the confinement is harmonic. What about using a more realistic harmonic oscillator basis along z? Would they expect any difference in the simulations?

WE RESPOND:
We have indeed used a plane wave basis in all directions, and hence $k$ is a three-dimensional vector. We have now specified this in the text.
Bradley et al. [J. Phys. B: At. Mol. Opt. Phys. 38, 4259 (2005)] have previously explored the effects of using a plane wave basis to represent a harmonically confined system. They suggest that, for a fixed number of grid points $N$, one can identify an optimal length $L$ of the simulation domain, for which the plane wave basis best matches the harmonic oscillator basis. In our case, this optimal domain size is calculated to be $L_z^\mathrm{opt} = \sqrt{2 \pi \hbar N_z / m \omega_z } \approx 11.5\mu m$ for our choice of $N_z=24$ grid points in the $z$-direction. The $z$-domain size in our simulations is $L_z = 16\mu m \approx 1.4 L_z^\mathrm{opt}$, i.e.~slightly larger than the optimal value. Despite this, the numerical density profile in the $z$-direction is well described by a Thomas-Fermi approximation, as shown in the attached figure (1). We therefore conclude that the $z$ density profile is well approximated by our choice of basis, and we would expect little difference if using a harmonic oscillator basis instead of plane waves.

THE REFEREE WRITES:
vi) Toward the end of section 4, the authors say that the above-cutoff atoms have been neglected in the calculation of the condensate fraction. Do they have any crude estimate of the effect of such an approximation? Related to this question, I see that they choose a cutoff energy of the same order of the chemical potential. Did they try to check what happens by changing it?

WE RESPOND:
We have revised the wording regarding the numerical condensate fraction. We now state that:
"...this parameter should be treated as a self-consistent measure of coherence within the c-field model, and may be different from the experimentally measured value."

Strictly speaking, it is an oversimplification to just add the "above-cutoff" atom number to the c-field atom number to match the experiment. In fact, the c-field atom number may already exceed the expected atom number, due to the different momentum distribution at high wavenumbers (see our response to point iv). It is therefore safer to simply consider the c-field condensate fraction as a self-consistent measure of coherence within the model, although we expect that most likely it would be within $\sim 0.1$ of the experimental value.

We have not tried numerically varying the cutoff, due to the computational cost of running additional simulations, as well as the difficulty in finding parameters that would provide a valid comparison between simulations with different cutoffs (we expand on this below).

When choosing the high energy cutoff, the fundamental requirement for validity is that $E_\mathrm{cut} \gtrsim \mu$, ensuring that the interacting modes are captured and that the plane wave basis is approximately diagonal at the cutoff (see e.g. Ref [23]). Adjusting the cutoff (while ensuring this requirement is still satisfied) would amount to changing the thermodynamic properties of the system, due to the inclusion or exclusion of high energy, noninteracting modes (occupied according to equipartition, as mentioned in our response to point iv). Observables like the condensate fraction would therefore increase or decrease. However, for any choice of (valid) cutoff there will always be some combination of SPGPE parameters $\lbrace g, \mu, T \rbrace$ that give rise to the same (average) equilibrium atom number and condensate fraction, and these states should also give approximately the same dynamics when evolved with the PGPE. However, it is not a priori clear what parameter values to use to keep these observables fixed as $E_\mathrm{cut}$ is varied, and hence systematically comparing results with different $E_\mathrm{cut}$ is a nontrivial task.

THE REFEREE WRITES
vii) Soon after they say "we find that 2D simulations provide quantitatively similar predictions for the decay of the imprinted excitation", but this sentence is somewhat cryptic. In a purely 2D gas in the same ring, the transition would be of BKT type and the condensate fraction would not behave in the same way. So, when comparing 2D with 3D, what is kept fixed? Condensate fraction? T/T$_\mathrm{\mathbf{c}}$? Number of atoms? As it is, this comment is not useful and might be a source of confusion.}

WE RESPOND:
We thank the referee for pointing this out. We agree that we should be more specific when comparing 2D and 3D results. Specifically, for a fair comparison we kept fixed (to within a few percent) the number of atoms and the condensate fraction between both calculations. The graph (Figure 2) attached shows the $Q$-factor as a function of the initial imprint amplitude $\alpha_0$ for a condensate fraction of $n_0 \approx 0.75$, showing close agreement between the two.

THE REFEREE WRITES:
viii) In the caption of fig.8, I guess $l=9$, not $l=1$. Also, I think it is better to write the values of both $m^*$ and $l$.}

WE RESPOND:
In the experimental sequence image Fig. 8(f), there is only one fringe, and hence $l=1$ (note that the data in Fig. 8 does not correspond to that in Fig. 9, where $l=9$). As suggested, we have added the value of $m^*$ in the caption of Fig. 8 for clarity.

Attachment:

Figures.pdf

---

## Round 1 · Referee Report · Anonymous (Referee 2) · 2023-2-27

Strengths

1 - Extensive numerical simulations and analysis 2 - Experimental observations well explained 3- Improvement of performance with respect to literature

Weaknesses

1 - Insufficient motivation as a calibration tool for more sensitive interferometers
2 - Insufficient description of the fitting procedure

Report

The manuscript "Viability of rotation sensing using phonon interferometry in Bose-Einstein condensates'' reports an experimental and numerical investigation of an interferometric gyroscope based on a Bose-Einstein condensate trapped in a ring potential.
The rotation frequency is sensed by measuring the rate of rotation of the angular standing wave initially imprinted on the condensate density by a suitable time-pulsed potential.

First the Authors investigate the sensitivity of the gyroscopes. In absence of any rotation, they characterize the persistence of the imprinted standing wave varying the number of nodes, the amplitude and the temperature of the sample. This characterization is performed numerically, by running simulations of the experiments with the formalism described that takes into account both the mean-field wavefunction of the condensate and the thermal excitations. The calculated column density is Fourier decomposed along the angular coordinate of the ring and the Fourier components are studied versus time. These are oscillating functions with exponentially decaying amplitudes, that can contain two time constants: the fast one inversely proportional to the initial, imprinted, deformation; the slow one nearly independent. The time-dependent amplitudes are obtained from the Fourier components divided by a sine function obtained with frequency and phase obtained from a fit.

After the characterization, an experimental part is reported. The interferometer should sense rotations, but, except for the earth rotation, the laboratory isn't rotating. Thus the Authors chose to impart a rotation to the superfluid, so that the condensate is at rest in a reference frame rotating with respect to the lab, and, vicecersa, the lab is rotating with respect to the condensate. Imparted rotations must have angular frequencies that are multiples of circulations quanta. The angular velocity are measured from the complex phase of the angular Fourier components, characterized earlier on. Using an alternative complementary technique, the angular velocity are also read from the interference pattern with a static condensate in a coaxial ring, occurring in time-of-flight, once all trapping potentials are removed.
The two methods agree within the error bar.

Finally, the sensitivity of the interferometer is discussed. It improves upon literature results but it still only of the order of 0.3 rad/s, an order of magnitude above the SQL set be shot-noise detection of the atomic densities.

The paper contains a good lot of material, especially a good lot of numerical simulations, supporting a convincing discussion about the decay mechanism of the phonons excited by the imprinted density deformation. In my opinion, this is the main strength of the manuscript.

The paper is well written and instructive. I recommend publication.

Requested changes

A few points could be improved: 1 - The analysis of the numerical data is based on the extraction of the time-dependent amplitude $\alpha (t)$. It is unclear what is the fitting function used to derive the sine function that divides the $m$-th Fourier component $A_m (t)$. 2- In Fig. 10, the caption says that the numerical simulation provide a precise control over the winding number $l$. What is the reason of the horizontal error bar in plot 10 (b)? Please explain. 3- The concluding statement

{\it "Significant improvements in sensitivity would be required to produce a sensor of comparable precision to those already commercially available. However, the superfluid’s absolute FoR does prevent measurement drift, potentially providing a very significant potential advantage for operation over extended periods. These devices may therefore prove useful if used in conjunction with conventional rotation sensors as a means of calibration to reduce uncertainty in classical systems over prolonged operation."}

How could this interferometer be used over prolonged operation, without being limited by the lifetime of the phonons excitation or the lifetime of the condensate? What would be, at least in principle, the protocol to use the interferometer as a means of calibration? I think this point deserves a better explanation, since it is presented as the work motivation.

  • validity: high
  • significance: high
  • originality: good
  • clarity: good
  • formatting: excellent
  • grammar: excellent

Author:  Charles Woffinden  on 2023-06-12  [id 3725]

(in reply to Report 2 on 2023-02-27)
Category:
answer to question

We thank the referee for their constructive feedback. We believe that the manuscript is now stronger as a result of the changes made to address these comments. Please find below our response to each point, including descriptions of any changes made to the manuscript. For reference, we have attached a copy of the manuscript with all changes highlighted in red.

THE REFEREE WRITES: 1 - The analysis of the numerical data is based on the extraction of the time-dependent amplitude $\alpha(t)$. It is unclear what is the fitting function used to derive the sine function that divides the $m$th Fourier component $A_m(t)$.

WE RESPOND: To clarify: throughout this work we have fitted the $A_m(t)$ data to Eq. (9), always assuming that the envelope function is exponentially decaying [i.e. assuming Eq. (10)]. All experimental data is consistent with this assumption, as well as most numerical data. The exceptions are:

  • For large enough amplitudes $\alpha_0$, in which case two exponential decays are visible (as explored in Fig 4 and the relevant text).
  • For low enough temperatures, in which case the envelope function is initially flat, before gradually curving towards what appears to be exponential decay (e.g. in Fig 6a).

Even in these cases we fit a decaying exponential to the data, although we vary the fitting window (as described in the text).

To avoid confusion, we have updated some of the text:

  • The fourth sentence of 6.2.2 has been changed from: "...we plot the time-varying envelope $\alpha(t)$, which we have isolated by dividing by a sinusoidal fit." to: "...we plot the time-varying envelope $\alpha(t)$, which we have isolated by dividing by the sinusoidal component of the fit, $\sin(\omega_{m^*}t + \phi_0)$,

with fitting parameters

$\omega_{m^*}$ and $\phi_0$"

  • We have removed the words "a sinusoidal fit" in the caption of Fig 4.
  • In the caption of Fig 6, we have replaced the words "a sinusoidal fit'' with "the sinusoidal component of the fit to Eqs.~(9)-(10)".

THE REFEREE WRITES: 2 - In Fig. 10, the caption says that the numerical simulation provide a precise control over the winding number $l$. What is the reason of the horizontal error bar in plot 10 (b)? Please explain.

WE RESPOND: In Sec.~7.3, we had previously written (regarding the experimental data): "The horizontal [error] bars are dominated by the variation in rotation frequency over the finite width of the ring due to the irrotational nature of the flow." This was in reference to the comment below Eq.~(12), where we point out that the rotation rate varies across the width of the ring. We have added "(see Sec.~7.1)" to the end of this sentence to make the link clearer. The horizontal error bars in Fig. 10(b) have the same origin (since the ring has the same dimensions in the simulations). We have added the following sentence in Sec 7.4 to clarify this: "However, the horizontal error bars remain approximately the same size as the experimental data, due to the aforementioned variation in rotation frequency over the width of the ring."

THE REFEREE WRITES: 3 - The concluding statement: How could this interferometer be used over prolonged operation, without being limited by the lifetime of the phonons excitation or the lifetime of the condensate? What would be, at least in principle, the protocol to use the interferometer as a means of calibration? I think this point deserves a better explanation, since it is presented as the work motivation.

WE RESPOND: We have added some further description at the end of the conclusion to describe how the the phonon rotation sensor could be used to counteract drift in a classical sensor during prolonged operation.

---

## Round 2 · Referee Report · Anonymous (Referee 3) · 2023-6-13

Report

The authors' response to my comments and the changes made to the manuscript are fully satisfactory. I recommend publication.

---

## Round 2 · Author Response

Dear Editor,

Thank you for your response and that of the referees. We have updated the manuscript to address the comments of both referees and look forward to receiving the reports on the updated version.

---

## Round 2 · List of Changes

We have updated Section 3.1 to state that the thermodynamics of the system are expected to be 3-D and the condensate fraction we achieve is the highest possible with our current experimental configuration.

We have made the red line in Figure 1 thicker to assist with visibility.

We have updated Section 4 to be make clearer that the model is not purely qualitative and added more detail around the cutoff method. We have also detailed further the conditions which remain fixed when comparing 2-D and 3-D simulations.

In Section 6.6.2, we have clarified the fitting function used to derive the sine function. This has also been clarified in the captions to Figure 4 and Figure 6.

In Sec. 7.3, we had previously written (regarding the experimental data): “The horizontal [error] bars
are dominated by the variation in rotation frequency over the finite width of the ring due to the irrotational nature of the flow.” This was in reference to the comment below Eq. (12), where we point
out that the rotation rate varies across the width of the ring. We have added “(see Sec. 7.1)” to the end of this sentence to make the link clearer.

In the caption of Figure 8, we have added the value of m*.

We have added the following sentence in Sec 7.4 to clarify origin of the horizontal error bar in Figure 10b.
“However, the horizontal error bars remain approximately the same size as the experimental data, due
to the aforementioned variation in rotation frequency over the width of the ring.”

We have added some further description at the end of the conclusion to describe how the the phonon
rotation sensor could be used to counteract drift in a classical sensor during prolonged operation.

---

## Editorial Decision

published